# Newer Therapies for Refractory *Helicobacter pylori* Infection in Adults: A Systematic Review

**DOI:** 10.3390/antibiotics13100965

**Published:** 2024-10-12

**Authors:** Ligang Liu, Milap C. Nahata

**Affiliations:** 1Institute of Therapeutic Innovations and Outcomes (ITIO), College of Pharmacy, The Ohio State University, Columbus, OH 43210, USA; liu.10645@osu.edu; 2College of Medicine, The Ohio State University, Columbus, OH 43210, USA

**Keywords:** *H. pylori* refractory infection, vonoprazan, rescue therapy, rifabutin, high-dose dual therapy

## Abstract

Background: *Helicobacter pylori* (*H. pylori*) infection is a global health concern, affecting approximately two-thirds of the world’s population. Standard first-line treatment regimens often fail, necessitating alternative rescue therapies. Objectives: This review aims to evaluate the efficacy and safety of newer treatment regimens in patients who have failed initial *H. pylori* eradication therapy. Methods: A comprehensive literature search was conducted in PubMed, the Cochrane Library, and Embase. Inclusion criteria were randomized controlled trials (RCTs) published after 2010, involving patients with previous *H. pylori* treatment failure and interventions with vonoprazan-based therapy, high-dose PPI–amoxicillin dual therapy (HDDT), or rifabutin-containing triple therapy. Results: 10 RCTs were included. HDDT demonstrated high eradication rates (81.3% to 89.2%), particularly when combined with metronidazole (92.6%), although at an increased frequency of adverse events. Vonoprazan-based regimens achieved comparable or higher eradication rates (83.3% to 89.5%) compared to PPI-based therapies, with similar adverse events. Rifabutin-containing triple therapy showed high efficacy (80.7% to 100%), particularly in patients with a history of multiple treatment failures, and it was associated with lower adverse events compared to bismuth-containing regimens. Conclusions: HDDT, vonoprazan-based therapy, and rifabutin-based therapy have proven to be effective and safe rescue regimens for treating *H. pylori* infection. Additional large-scale randomized studies are needed to determine the optimal doses and durations of these regimens to achieve the highest eradication rate with the lowest incidence of adverse events among patients with refractory *H. pylori* infections globally.

## 1. Introduction

*Helicobacter pylori* (*H. pylori*) infection affects about two-thirds of the population worldwide and leads to significant morbidity and mortality [1,2]. It is the most common cause of peptic ulcer disease and is strongly linked to chronic gastritis and gastric cancer [3,4]. *H. pylori* is responsible for almost 90% of distal gastric cancers, with the lifetime gastric cancer risk for infected individuals in the range of 1–5% [4,5]. Moreover, it may also exacerbate conditions such as vitamin B12 deficiency, metabolic syndrome, diabetes, and non-alcoholic liver disease [6].

*H. pylori* targets the stomach lining [7] and is adept at surviving in the stomach’s acidic environment by burrowing into the mucous layer, causing inflammation and damage [8,9,10]. The risk factors for *H. pylori* infection include lower socioeconomic status, poor hygiene, dining at restaurants, alcohol consumption, and smoking [11,12,13]. Additionally, infection may be associated with ethnicity, with Black and Hispanic populations showing higher infection rates compared to Whites [14]. *H. pylori* infection is typically diagnosed via urea breath tests and stool antigen tests [15]. Eradication therapy is recommended for all infected individuals, with particular emphasis on testing and treating high-risk populations. This approach has been shown to reduce gastric cancer risk, especially in those with a family history of gastric cancer [16,17].

Treatment guidelines generally highlight the importance of achieving eradication on the first attempt to avoid the need for retreatment and the development of antibiotic resistance [18]. Traditional treatment has frequently comprised a combination of antibiotics and a proton pump inhibitor (PPI) [19]. An effective treatment regimen should achieve an eradication rate of over 90% [20]. Recommended first-line therapies for *H. pylori* infection have included clarithromycin triple therapy, bismuth quadruple therapy, concomitant therapy, and sequential therapy [19,21]. Clarithromycin triple therapy includes the use of a PPI, clarithromycin, and either amoxicillin or metronidazole, administered for 14 days. Bismuth quadruple therapy (BQT) consists of a PPI, bismuth, tetracycline, and nitroimidazole (such as metronidazole or tinidazole), with a treatment duration of 10 to 14 days. Concomitant therapy has included a PPI, clarithromycin, amoxicillin, and nitroimidazole taken for 10 to 14 days. Sequential therapy has offered a two-phase treatment: initially, a PPI and amoxicillin given for five to seven days, followed by a PPI, clarithromycin, and nitroimidazole for an additional five to seven days [19,21].

When initial therapy has failed, bismuth quadruple therapy, levofloxacin salvage regimens, and clarithromycin-containing regimens have been the preferred options based on local antimicrobial resistance rate and patient’s previous exposure to antibiotics [19,22]. However, increased antibiotic resistance and patient nonadherence due to adverse events of medications have made current treatment regimens inadequate to achieve desired outcomes [23]. *H. pylori* resistance rates were shown to be 21.4% for clarithromycin, 15.8% for levofloxacin, and 38.9% for metronidazole in Europe. The resistance rates in the Asia–Pacific region were shown to be 22% for clarithromycin, 52% for metronidazole, 26% for levofloxacin, 4% for tetracycline, and 4% for amoxicillin [24,25]. Therefore, the search for effective and safe treatment regimens for *H. pylori* infection in patients who have failed initial therapy has become a high priority.

Newer regimens, such as high-dose PPI–amoxicillin dual therapy (HDDT), vonoprazan-based therapy, and rifabutin-containing triple therapy, offer alternative mechanisms of action and improved acid suppression, providing potential advantages over traditional rescue regimens. Vonoprazan, a potassium-competitive acid blocker (P-CAB), has shown a more profound effect on gastric acid suppression compared to PPIs [26]. Results from clinical trials and meta-analyses have proven the effectiveness and safety of vonoprazan-based therapy as a first-line treatment in treating *H. pylori* infections [27,28,29], providing evidence for the use of vonoprazan in patients who have failed initial PPI-based therapy. Rifabutin acts by inhibiting the β-subunit of bacterial DNA-dependent RNA polymerase, resulting in bactericidal activity [30]. Rifabutin-containing triple therapy has also demonstrated high efficacy with few adverse events and high treatment adherence [31]. Furthermore, high-dose PPI combined with amoxicillin dual therapy has also proven to be effective and safe as a non-first-line therapy [32].

This review aims to describe the efficacy and safety of newer treatment regimens for *H. pylori* infection in patients who have failed first-line therapy and to provide guidance for the use of various therapies in clinical practice.

## 2. Methods

A literature review was conducted using databases such as PubMed, the Cochrane Library, and Embase on 9 February 2024. The search terms included “*Helicobacter pylori*”, “*H. pylori*”, second-line, third-line, fourth-line, rescue, salvage, refractory, and RCT*, random*, “randomized controlled trial*”, intervention, trial* or “clinical trial*”. The inclusion criteria were as follows: (1) patients with previous treatment failure; (2) interventions involving vonoprazan-based therapy, PPI–amoxicillin dual therapy, or rifabutin-containing therapy; (3) randomized controlled trials; (4) reported results on eradication rate; and (5) published after 2010. Studies conducted in other countries outside the US were also included. Studies such as reviews, meta-analyses, posters, meeting abstracts, and case reports were excluded. Only full-text studies were included.

## 3. Results

The initial search identified 1010 articles; 313 articles were included in PubMed, 275 in Embase, and 422 in the Cochrane Library. A total of 385 duplicate documents were excluded. After a preliminary screening of the titles or abstracts, 505 papers were excluded. Fifty articles were excluded after a review of the published abstracts. After carefully reading the full texts, an additional 37 articles were excluded. Ultimately, 10 studies were selected for inclusion [31,32,33,34,35,36,37,38,39,40] (Figure 1). Key characteristics of the included studies, the medication regimens, eradication rates, and the adverse events rate can be found in Table 1.

### 3.1. High-Dose PPI–Amoxicillin Dual Therapy (HDDT)

Five studies explored the efficacy of HDDT compared to other well-established rescue therapies, with some employing modifications to enhance their effectiveness [32,33,34,35,36]. Bi et al. [32] conducted a multicenter, randomized controlled trial to evaluate the efficacy and safety of HDDT compared to bismuth-containing quadruple therapy in patients with prior *H. pylori* treatment failures. The HDDT regimen achieved eradication rates of 75.4% in the intention-to-treat (ITT) analysis and 81.3% in the per protocol (PP) analysis. Moreover, patients in the HDDT group experienced significantly fewer adverse events compared to the bismuth-containing quadruple therapy group (11.1% versus 26.8%, *p* < 0.001). Yang et al. [33] compared HDDT against traditional alternative rescue regimens and found that the HDDT group achieved higher eradication rates (89.3%) compared to sequential therapy (51.8%) and levofloxacin-containing triple therapy (78.6%) in ITT analysis. The overall occurrence of adverse events was similar among all groups. Okimoto et al. [36] observed that the eradication rates were 73.9% in the levofloxacin triple therapy group and 64.0% in the HDDT group in the ITT analysis, with no significant differences in adverse events between the two groups.

Some researchers explored modifications to HDDT to enhance its efficacy [34,35]. Ding et al. [34] found that the addition of metronidazole to HDDT significantly improved eradication rates, reaching 85.8% in ITT and 92.6% in PP analysis, compared to 73.1% and 83.1% for HDDT alone, respectively. However, a higher incidence of adverse events was observed with the metronidazole-containing regimen (23.1% vs. 6.0%, *p* < 0.001). In contrast, Goh et al. [35] found no significant improvement in eradication rates when bismuth was added to HDDT (80.5% ITT and 82.5% PP) compared to HDDT alone (84.6% ITT and 89.2% PP), with similar adverse effect profiles.

The most frequently reported adverse events for HDDT included nausea, diarrhea, dysgeusia, abdominal pain, flatulence, headache, dizziness, decreased appetite, constipation, fatigue, and skin rash [32,33,34,35,36]. Only a few participants experienced severe adverse events that led to treatment discontinuation. Most adverse events were mild-to-moderate, and patients typically recovered after treatment [32,33,34,35,36].

### 3.2. Vonoprazan-Based Therapy

Two RCTs reported the efficacy and safety of vonoprazan-based therapy versus PPI-based therapy for patients who had failed first-line therapy [37,38]. Hojo et al. [37] found that the eradication rates in the vonoprazan and PPI groups were 73.9% vs. 82.6% in ITT and 89.5% vs. 86.4% in PP analyses, with no significant differences in the occurrence of adverse events between the groups. Sue et al. [38] noticed that vonoprazan-based therapy achieved higher eradication rates with PP analysis (83.3%) compared to PPI-based therapy (57.1%, *p* = 0.043), although the differences with ITT analysis were not statistically significant. Both studies reported that adverse events from vonoprazan-based therapy were generally mild-to-moderate, and few participants discontinued treatment. The most common adverse events included nausea, diarrhea, dysgeusia, abdominal discomfort, abdominal bloating, and belching [37,38].

### 3.3. Rifabutin-Containing Triple Therapy

Three RCTs studied the use of rifabutin as a rescue therapy in patients with previous treatment failures [31,39,40]. Chen et al. [31] compared rifabutin-containing triple therapy (150 mg twice daily) with bismuth quadruple therapy in a large-scale multicenter trial in China. Both therapies demonstrated high eradication rates (89.0% for rifabutin vs. 89.6% for bismuth) in ITT analysis, and the rifabutin group showing significantly lower adverse events (26.4% vs. 54.4%, *p* < 0.001). Lim et al. [39] evaluated rifabutin-based triple therapy with 30 mg lansoprazole versus 60 mg lansoprazole. The higher dose of lansoprazole achieved eradication rates of 96.3% (ITT) and 100% (PP), significantly higher than the lower-dose group (78.1% ITT, 80.6% PP, *p* = 0.047). The treatment was well tolerated, with mild adverse effects. Mori et al. [40] assessed 10-day and 14-day rifabutin-based therapies (300 mg once daily) in a small group of Japanese patients and found that the eradication rates were 83.3% and 94.1% in ITT analyses, respectively. However, the incidence of adverse events was high (83.3% to 94.1%) but comparable between the two durations of therapy. Across the three studies, adverse events from rifabutin-containing triple therapy were generally mild and manageable. The most commonly observed adverse events were diarrhea, fever, skin rash, myalgia, and fatigue. The treatments were well tolerated, and all adverse events were reversible [31,39,40].

## 4. Discussion

This review has described the efficacy and safety of newer treatment regimens for *H. pylori* eradication in patients who have failed initial therapy. HDDT, vonoprazan-based regimens, and rifabutin-containing triple therapy demonstrated high eradication rates, offering viable options for patients where first-line treatments have failed. HDDT, especially when combined with metronidazole, achieved high eradication rates, albeit with a higher rate of adverse events. Vonoprazan-based therapies showed a comparable or higher eradication rate compared to PPI-based regimens. Rifabutin-containing therapy emerged as a third-line option with high eradication rates and a lower incidence of adverse effects compared to bismuth-containing regimens.

*H. pylori* infection is a major global health concern. Effective eradication on the first attempt is critical, as failure can lead to increased treatment resistance, requiring the use of more complex treatment regimens and greater healthcare costs [41]. Failure of first-line *H. pylori* eradication therapy is not uncommon, with global eradication rates often falling below the desired threshold of 90% [42]. Approximately 20% to 30% of patients fail first-line therapy in the US [43]. This failure is primarily driven by increasing antibiotic resistance, poor patient adherence, and suboptimal treatment regimens [23,44]. Treatment of patients after initial therapy failure presents a significant clinical challenge. The need for effective and tolerable rescue regimens is crucial in managing refractory *H. pylori* infections. The newer regimens explored in this review (HDDT, vonoprazan-based therapies, and rifabutin-containing therapies) address key gaps in the current treatment strategies for *H. pylori* infection. These regimens offer high eradication rates, and some of them were shown to reduce adverse effects compared to traditional treatments, such as bismuth-containing therapies.

The effectiveness of these newer regimens lies in their ability to overcome the limitations of traditional therapies through enhanced acid suppression, alternative mechanisms of bacterial inhibition, and the strategic use of antibiotics that are less prone to development of resistance. By optimizing the gastric environment and employing antibiotics with distinct mechanisms, these therapies provide a higher likelihood of achieving eradication in patients where standard regimens have failed. An HDDT regimen enhances the efficacy of amoxicillin by maintaining elevated gastric pH levels, which optimizes the antimicrobial activity against *H. pylori* [45]. Vonoprazan provides more potent and sustained gastric acid suppression than traditional PPIs, creating a more favorable environment for antibiotics to kill *H. pylori* [46]. Rifabutin targets bacterial RNA polymerase and bypasses common resistance pathways, highlighting its role as a potent salvage therapy [47].

The use of metronidazole and rifabutin in *H. pylori* rescue therapy, while effective, may present a significant challenge due to elevated adverse events. Metronidazole is known to cause gastrointestinal discomfort, including nausea, metallic taste, and antibiotic-associated diarrhea [48]. The rate of adverse events of metronidazole-containing regimens can be as high as 47.8% [37]. Similarly, rifabutin can cause gastrointestinal disturbances, myalgia, and uveitis [49]. The adverse events of rifabutin appear to be dose-dependent and may vary across populations. A large Chinese study reported significantly lower adverse event rates for rifabutin (150 mg twice daily) compared to bismuth-containing quadruple therapy [31]. Conversely, a small Japanese study observed a high adverse event rate of 94.1% when rifabutin was administered at 300 mg once daily [40]. These findings suggest that a dosage regimen of 150 mg twice daily might be safer while maintaining efficacy.

The occurrence of adverse events can decrease treatment adherence and patient quality of life. The European Registry’s study on *H. pylori* management found that 1.7% of patients were nonadherent to treatment regimens, with higher rates seen among those prescribed treatment regimens with a longer duration, rescue therapies, and those who experienced adverse events [50]. A recent study found that about 23% of patients experienced at least one adverse event related to *H. pylori* eradication treatments, such as taste disturbance (7%), diarrhea (7%), nausea (6%), and abdominal pain (3%) [51]. Mitigating adverse events during treatment involves several strategies. Conducting pre-treatment assessments, such as evaluating for history of antibiotics allergies or drug intolerances, can prevent complications from *Helicobacter pylori* treatment. Clinicians should educate patients about potential adverse effects, such as nausea, diarrhea, dysgeusia, and fatigue, encourage prompt reporting for timely management, and emphasize the importance of medication adherence to prevent antibiotic resistance and treatment failure. Enhanced monitoring and educational interventions, such as telephone follow-up re-education, short-message services, and smart phone applications could be employed as well to enhance the treatment adherence. Evidence shows that enhanced educational interventions have had positive effects on both the *H. pylori* eradication rate and adherence [52]. Another practical approach is the potential use of probiotics or prebiotics, which may restore the gut microbiota balance and reduce gastrointestinal adverse effects [53,54]. In addition, lowering the dose of metronidazole can reduce the risk of adverse events while maintaining efficacy [55]. Shorter treatment durations, when clinically appropriate, can further decrease cumulative drug exposure and the likelihood of adverse effects. Personalized therapy based on regional resistance profiles may also enhance the optimization of medication regimens to achieve the desired efficacy of antibiotics [56].

Recommendations for treatment

Our recommendations for managing refractory *H. pylori* infections, addressing patient-specific factors such as prior treatment failures and penicillin allergies, are shown in Figure 2. For patients who have failed clarithromycin-based regimens as an initial therapy, BQT might be recommended as a first-line rescue therapy if the patients have a penicillin allergy. For patients without a penicillin allergy, the treatment regimens could be either HDDT or BQT. Given the lower incidence of adverse events associated with HDDT compared to BQT, HDDT might be the preferred option, as it may result in better tolerability while maintaining high eradication rates. If the first-line rescue therapy fails, vonoprazan-based therapy with metronidazole or levofloxacin-based therapy should be considered as second-line rescue therapies. Third-line rescue treatments may include vonoprazan-based therapy with sitafloxacin or rifabutin-based therapy.

For patients who have failed BQT as an initial regimen, several next-step treatments are available. For patients with a penicillin allergy, repeating BQT with high-dose PPI or switching to levofloxacin-based triple therapy are effective first-line rescue options. In patients without a penicillin allergy, HDDT should be the preferred treatment for a first-line rescue regimen. For second-line rescue therapies, vonoprazan-based therapy could be used with metronidazole as an option, or levofloxacin-based therapy can be used if needed. Third-line therapies, such as rifabutin-containing triple therapy and vonoprazan-based therapy with sitafloxacin, can be used in those who have not responded to first- and second-line rescue therapies.

### Limitations

First, most studies were conducted in China and Japan, which may restrict the applicability of the results to other populations residing in other countries with different antibiotic resistance patterns. These findings do not fully represent the global population due to lack of adequate studies, especially in Western countries. Another limitation is the small sample sizes of most of the included trials. The small sample sizes limit the generalizability of the results and make it difficult to draw robust conclusions. The absence of diverse ethnic and geographic representation in the study populations restricts the relevance of the findings to global clinical practice.

Future research should focus on identifying optimal regimens to mitigate adverse effects, such as the use of lower doses, shorter treatment durations, or combining agents with complementary mechanisms that minimize toxicity while achieving the desired efficacy.

Future research should expand the geographical scope of studies to include diverse populations from various regions with different resistance patterns. This is important, as the current data have largely come from studies conducted in regions like China and Japan. Larger multicenter randomized controlled trials are needed to provide additional robust data on the efficacy and safety of newer regimens. Moreover, future research should focus on thoroughly assessing the safety of different medications by providing detailed reports on adverse events, including their severity, duration, and management strategies. Such comprehensive data will enable clinicians to better anticipate potential adverse effects and implement effective strategies to mitigate them, which, in turn, could enhance patient adherence and improve overall treatment outcomes. Furthermore, future research should evaluate optimal regimens to mitigate adverse effects, such as the use of lower doses of antibiotics, shorter treatment durations, or combined medications with different mechanisms. One possible approach may involve investigating antibiotic combinations that utilize synergistic effects to lower the overall required dose, thereby reducing potential toxicity. For example, dual-therapy regimens involving PPI with amoxicillin may be optimized by adjusting their ratios to achieve efficacy at lower doses. Administration of PPI–amoxicillin dual therapy four times daily has shown greater efficacy and safety compared to other regimens [57]. Shortening the treatment duration might also mitigate adverse events, as longer regimens are associated with higher incidences of adverse effects [58]. Additionally, combining antibiotics with probiotics could reduce the risk of gastrointestinal disturbances while maintaining the desired eradication rates [59]. Future research should also emphasize antibiotic susceptibility testing or genotypic resistance-guided therapy to personalize treatment regimens, including the assessments of their cost-effectiveness among patients [60,61,62]. Additionally, an expansion of global and regional databases of resistance patterns and data regarding clinical efficacy and safety outcomes could guide the location-specific selection of treatments, improving eradication success in diverse populations [63].

## 5. Conclusions

High-dose PPI and amoxicillin dual therapy, vonoprazan-based therapy, and rifabutin-based therapy have proven to be effective and safe rescue regimens for treating *H. pylori* infection. These regimens have generally achieved eradication rates exceeding 80%, with some studies reporting rates approaching 90%. The incidence of adverse events associated with these treatments was comparable to or lower than traditional rescue regimens, such as PPI-based triple therapy or bismuth quadruple therapy. Further well-designed large-scale randomized studies are required to determine the optimal doses and durations of various treatment regimens to achieve the highest eradication rate with the lowest incidence of adverse events among patients with refractory *H. pylori* infections.

## Figures and Tables

**Figure 1 antibiotics-13-00965-f001:**
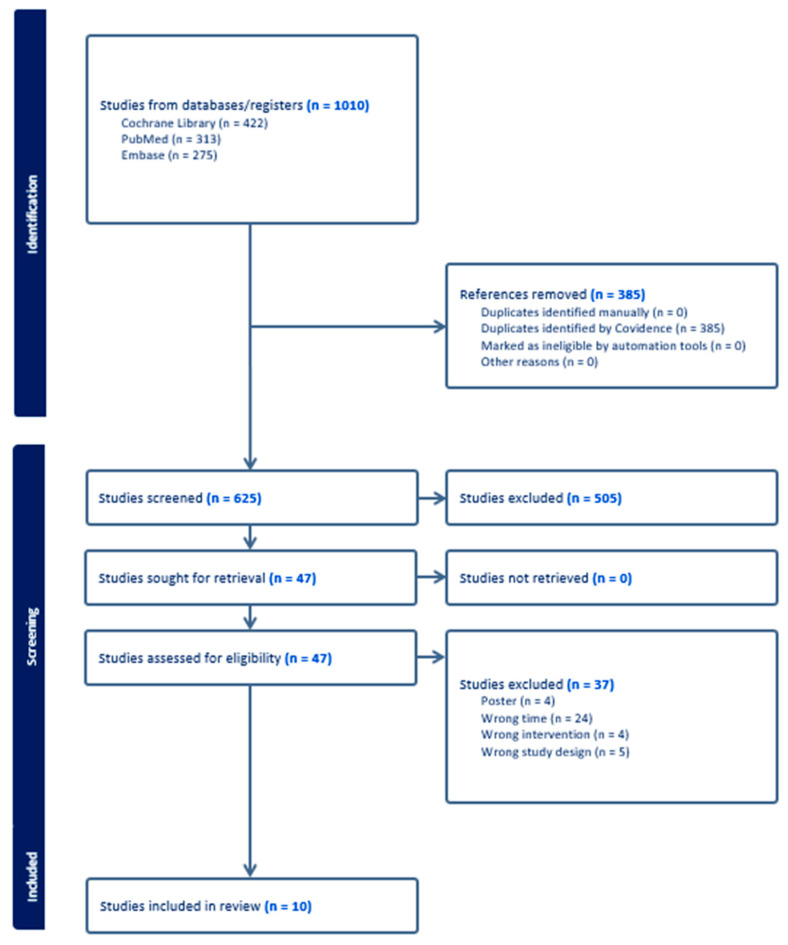
Study selection flowchart.

**Figure 2 antibiotics-13-00965-f002:**
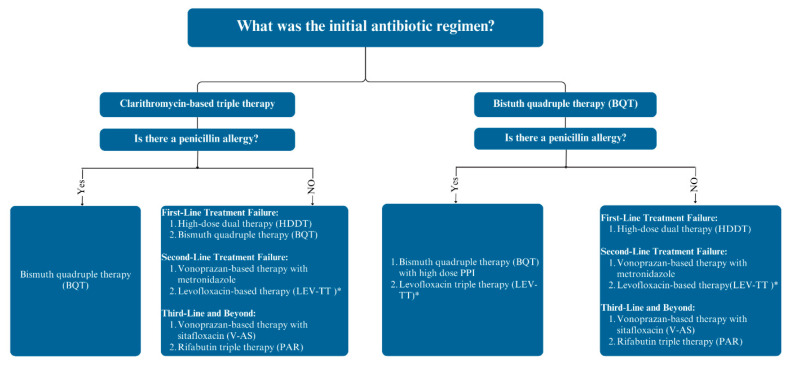
Approach to antibiotic treatment in patients with persistent *Helicobacter pylori* infection. Abbreviation: PPI, proton pump inhibitor. * If the local levofloxacin resistance rates were less than 15%.

**Table 1 antibiotics-13-00965-t001:** Summary of included studies and the efficacy and safety results.

Author, Year	Country	Study Design	Population	Patient Sample Size (N)	Intervention Treatment Regimen	Control Treatment Regimen	Duration of Therapy	Successful Eradication Time	Eradication Rate for Therapies %		Adverse Events %
High-dose PPI–Amoxicillin dual therapy (HDDT)									PP	ITT	
Bi, 2022 [32]	China	Prospective, randomized, multicenter, non-inferiority trial	At least one previous failure of *H. pylori* eradication	658	HDDT (N = 329): ESO 40 mg and AMO 1000 mg TID	BQT (N = 329): ESO 40 mg BID, bismuth 220 mg BID, tetracycline 500 mg TID, and furazolidone 100 mg BID	14 days	^13^C/^14^C-UBT or HpSAT at 4 to 8 weeks following eradication treatment	HDDT: 81.3%	HDDT: 75.4%	HDDT: 11.1%
									BQT: 85.1%	BQT: 78.1%	BQT: 26.8%
Ding 2023 [34]	China	Prospective, randomized, open-label, parallel-controlled, single-center superior clinical trial	Two or more previous failures of *H. pylori* eradication	268	HDDT-M (N = 134): ESO 40 mg BID, AMO 1000 mg TID and MNZ 400 mg TID	HDDT (N = 134): ESO 40 mg twice daily and AMO 1000 mg TID	14 days	Urea breath test at 6 weeks	HDDT+M: 92.6%	HDDT+M: 85.8%	HDDT+M: 23.1%
									HDDT: 83.1%	HDDT: 73.1%	HDDT: 6.0%
Goh, 2020 [35]	Japan	Comparative, randomized, open-label study	Failed first-line therapy	80	HDDT-Bi (N = 41): dexlansoprazole 60 mg BID and AMO 1 g QID, with bismuth 240 mg BID	HDDT (N = 39): dexlansoprazole 60 mg BID and AMO 1 g 4 QID	14 days	^13^C-UBT at least 4 to 8 weeks after they completed the treatment.	HDDT-Bi: 82.5%	HDDT-Bi: 80.5%	NA, but no difference
									HDDT: 89.2%	HDDT: 84.6%	
Yang 2015 [33]	Taiwan	Prospective, randomized study	Previously received anti-*H. pylori* therapies	168	HDDT (N = 56): RPZ 20 mg QID and AMO 750 mg QID for 14 days	ST (N = 56): RPZ 20 mg BID and AMO 1000 mg BID for 5 days, followed by RPZ 20 mg BID, MTZ 500 mg BID, and CLA 500 mg BID for 5 days	14 days vs 10 days vs 7 days	UBT at four to eight weeks after treatment completion	HDDT: 89.3%	HDDT: 89.3%	HDDT: 23.0%
						LEV-TT (N = 56): rabeprazole 20 mg BID, AMO 1000 mg BID, and LEV 250 mg BID for 7 days			ST: 53.7%	ST: 51.8%	ST: 33.2%
									LEV-TT: 78.6%	LEV-TT: 78.6%	LEV-TT: 26.8%
Okimoto, 2014 [36]	Japan	Prospective, randomized, controlled study	Failed Japanese first-line and second-line eradication therapy	51	HDDT (N = 27): RPZ 10 mg QID and AMO 500 mg QID for 14 days	LEV-TT (N = 24): RPZ 10 mg BID, AMO 750 mg BID, and LEV 500 mg QD for 10 days	10 vs. 14 days	UBT at 6 to 12 weeks after treatment completion	HDDT-L: 73.9%	HDDT-L: 73.9%	HDDT-L: 25%
									LEV-TT: 76.2%	LEV-TT: 64.0%	LEV-TT: 25%
Vonoprazan-based therapy											
Sue, 2019 [38]	Japan	Prospective, open label, randomized	Failed Japanese first-line and second-line eradication therapy	63	V-AS (N = 33): VPZ 20 mg BID, AMO 750 mg BID, and STFX 100 mg BID	P-AS (N = 30): LPZ 30 mg, RPZ 10 mg, or ESO 20 mg BID; AMO 750 mg BID; and STFX 100 mg BID	7 days	UBT	V-AS: 83.3%	V-AS: 75.8%	No difference
									P-AS: 57.1%	P-AS: 53.3%	
Hojo, 2020 [37]	Japan	Randomized, open-label, parallel-group study	Failed first-line eradication therapy	46	V-AM (N = 23): VPZ 20 mg, AMO 750 mg, and MNZ 250 mg BID	P-AM (N = 23): RPZ 10 mg, AMO 750 mg, and MNZ 250 mg BID	7 days	^13^C-urea breath test at least 4 weeks after the end of treatment	V-AM: 89.5%	V-AM: 73.9%	V-AM: 47.8%
									P-AM: 86.4%	P-AM: 82.6%	P-AM: 30.4%
Rifabutin-containing triple therapy											
Chen, 2023 [31]	China	Noninferiority, open-label, randomized trial	Previously failed 2 or more eradication regimens	364	PAR (N = 182): ESO 20 mg, AMO 1.0 g, and rifabutin 150 mg BID.	BQT (N = 182): ESO 20 mg and bismuth 600 mg BID, in addition to MTZ 400 mg and tetracycline 500 mg QID	14 days	UBT at 6 weeks after treatment completion	P-AR: 94.0%	P-AR: 89.0%	P-AR: 26.4%
									BQT: 95.3%	BQT: 89.6%	BQT: 54.4%
Lim, 2014 [39]	South Korea	Single-centered, randomized, open-label, and controlled clinical trial	Patients with two previous eradication failures	59	HD-PAR (N = 32): LAN 60 mg BID, AMO 1.0 g TID and rifabutin 150 mg BID	PAR (N = 27): LAN 30 mg BID, AMO 1.0 g TID and rifabutin 150 mg BID	7 days	^14^C-urea breath test 4 weeks after the therapy.	HD-PAR: 100%	HD-PAR: 96.3%	NA but no difference
									PAR: 80.7%	PAR: 78.1%	
Mori, 2016 [40]	Japan	Prospective, randomized, open-label study	Failed Japanese first-line and second-line eradication therapy	29	10-PAR (N = 12): ESO 20 mg QID, AMO 500 mg, QID, and rifabutin 300 mg QD	14-PAR (N = 17): ESO 20 mg QID, AMO 500 mg QID, and rifabutin 300 mg QD	10 vs. 14 days	UBT or HpSA at twelve weeks after the end of eradication therapy	10-PAR: 81.8%	10-PAR: 83.3%	10-PAR: 75.0%
									14-PAR: 91.7%	14-PAR: 94.1%	14-PAR: 94.1%

Abbreviations: HDDT, high-dose PPI–amoxicillin dual therapy; P-AS, PPI–amoxicillin–sitafloxacin; V-AS, vonoprazan–amoxicillin–sitafloxacin; P-AM, PPI–amoxicillin–metronidazole; V-AM, vonoprazan–amoxicillin–metronidazole; PAR, PPI–amoxicillin–rifabutin; BQT, bismuth quadruple therapy; 10-PAR, PPI–amoxicillin–rifabutin for 10 days; 14-PAR, PPI–amoxicillin–rifabutin for 14 days; HD-PAR, high dose PPI–amoxicillin–rifabutin; HDDT-Bi, high-dose PPI–amoxicillin dual therapy–bismuth; HDDT-M, high-dose PPI–amoxicillin dual therapy–metronidazole; ST, sequential therapy, LEV-TT, levofloxacin-containing triple therapy; PPI, proton pump inhibitor; ESO, esomeprazole; LPZ, lansoprazole; OME, omeprazole; RPZ, rabeprazole; VPZ, vonoprazan; AMO, amoxicillin; CLR, clarithromycin; MNZ, metronidazole, STFX, sitafloxacin; LEV, levofloxacin; QD, once daily; BID, twice daily; TID, three time daily; QID, four times daily; RCT, randomized controlled trial; UBT, urea breath test; HpSA, stool antigen tests.

## Data Availability

Not applicable.

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
