# Peer review of "Newer Therapies for Refractory Helicobacter pylori Infection in Adults: A Systematic Review"

_antibiotics, 2024, doi:10.3390/antibiotics13100965_

Round 1
Reviewer 1 Report
Comments and Suggestions for Authors
I think the article is well written. The authors took into account 10 trial articles published after 2010. They presented 3 new Helicobacter pylori therapies, namely High-Dose PPI-Amoxicillin Dual Therapy (HDDT), Vonoprazan-Based Therapy and Rifabutin-Containing Triple Therapy. In Table 1 they also showed the eradication rate and adverse events for each of the therapies.
I suggest 2 corrections:
1. In the Introduction the authors wrote that there is increasing antibiotic resistance. However, they did not provide details. It is necessary to add what Helicobacter pylori resistance looks like to currently used drugs with percentages, e.g. based on https://pubmed.ncbi.nlm.nih.gov/33837118/
2. In Table 1, it is also necessary to add in Abbreviations what P-AS, P-AM, P-AR, BQT, 10-PAR and others mean.
Comments on the Quality of English LanguageOK
Author Response
1. In the Introduction the authors wrote that there is increasing antibiotic resistance. However, they did not provide details. It is necessary to add what Helicobacter pylori resistance looks like to currently used drugs with percentages, e.g. based on https://pubmed.ncbi.nlm.nih.gov/33837118/
Thank you for your comments. We have revised the manuscript to include the details regarding Helicobacter pylori antibiotic resistance. The revision now incorporates specific resistance rates for currently used medications (Lines 66-69).
2. In Table 1, it is also necessary to add in Abbreviations what P-AS, P-AM, P-AR, BQT, 10-PAR and others mean.
Thank you for your suggestions. We have addressed your comment by adding the necessary abbreviations to Table 1. These additions should now clarify the terminology used within the Table and enhance the overall understanding of the data presented.
Reviewer 2 Report
Comments and Suggestions for Authors
1. The majority of the studies reviewed were conducted in China and Japan, which restricts the generalizability of the findings. The antibiotic resistance patterns and patient demographics in these regions may differ significantly from other parts of the world, limiting the applicability of the results to a broader, global population.
2. While the article highlights antibiotic resistance as a significant challenge in H. pylori treatment, it does not thoroughly discuss how specific resistance patterns influence the outcomes of the different regimens. Including more detailed information on resistance data from diverse regions would strengthen the case for personalized treatment approaches.
3. Although adverse events are mentioned, the analysis lacks depth. The severity, duration, and management of these adverse effects are not thoroughly explored, which limits the practical application of the findings. The authors should conduct a more detailed exploration of how to mitigate adverse events, would enhance the article's clinical relevance.
4. The article provides a good overview of various treatment regimens but falls short of offering concrete, practical guidance for clinicians. Authors need to provide more detailed recommendations on which regimens to choose in specific clinical scenarios would make the review more useful in real-world practice.
5. While the conclusion calls for more large-scale randomized controlled trials, it does not elaborate on the specific areas where future research should focus, such as optimal dosing, personalized therapies, or regional resistance patterns. This limits the forward-looking utility of the review.
Author Response
- The majority of the studies reviewed were conducted in China and Japan, which restricts the generalizability of the findings. The antibiotic resistance patterns and patient demographics in these regions may differ significantly from other parts of the world, limiting the applicability of the results to a broader, global population.
Thank you for your valuable feedback. We acknowledge that most of the studies reviewed were conducted in China and Japan. We have highlighted this as a limitation of our review. We also included statements that future research should aim to include a diverse set of regions to increase the global generalizability of the findings (Lines 306-309; 318-320).
- While the article highlights antibiotic resistance as a significant challenge in H. pylori treatment, it does not thoroughly discuss how specific resistance patterns influence the outcomes of the different regimens. Including more detailed information on resistance data from diverse regions would strengthen the case for personalized treatment approaches.
Thank you for your thoughtful comment. We have added specific resistance patterns from different regions to provide a clearer understanding of the regional variations in antibiotic resistance (Lines 66-69).
- Although adverse events are mentioned, the analysis lacks depth. The severity, duration, and management of these adverse effects are not thoroughly explored, which limits the practical application of the findings. The authors should conduct a more detailed exploration of how to mitigate adverse events, would enhance the article's clinical relevance.
Thank you for the constructive feedback. We acknowledge the importance of a thorough analysis of adverse events for practical clinical application. We have expanded our Results on the severity and duration of adverse effects associated with different treatment regimens and the potential strategies for their management (Lines 169-173;182-285;200-203;252-276).
- The article provides a good overview of various treatment regimens but falls short of offering concrete, practical guidance for clinicians. Authors need to provide more detailed recommendations on which regimens to choose in specific clinical scenarios would make the review more useful in real-world practice.
Thank you for your insightful comment. In response, we have expanded the Discussion section to include specific clinical scenarios and guidance on regimen selection (Lines 278-304).
- While the conclusion calls for more large-scale randomized controlled trials, it does not elaborate on the specific areas where future research should focus, such as optimal dosing, personalized therapies, or regional resistance patterns. This limits the forward-looking utility of the review.
Thank you for your suggestion. We have substantially revised the Discussion section to emphasize the need for additional randomized controlled trials that focus on optimal dosing strategies, the development of personalized therapies, and an examination of regional antibiotic resistance patterns (Lines 324-343).